# Prediction of Concrete Fragments Amount and Travel Distance under Impact Loading Using Deep Neural Network and Gradient Boosting Method

**DOI:** 10.3390/ma15031045

**Published:** 2022-01-28

**Authors:** Kyeongjin Kim, WooSeok Kim, Junwon Seo, Yoseok Jeong, Meeju Lee, Jaeha Lee

**Affiliations:** 1Major of Civil Engineering, National Korea Maritime & Ocean University, 727 Taejong-ro, Yeongdo-gu, Busan 49112, Korea; kkj4159@naver.com; 2Department of Civil Engineering, Chungnam National University, 99 Daehak-ro, Yuseong-gu, Daejeon 34134, Korea; wooseok@cnu.ac.kr; 3Department of Civil and Environmental Engineering, South Dakota State University, Brookings, SD 57007, USA; junwon.seo@sdstate.edu; 4Department of Construction and Disaster Prevention Engineering, Kyungpook National University, 2559 Kyungsangdae-ro, Sangju 37224, Gyeongsangbuk-do, Korea; ysjeong@knu.ac.kr; 5Major of Civil and Environmental Engineering, National Korea Maritime & Ocean University, 727 Taejong-ro, Yeongdo-gu, Busan 49112, Korea; 6Interdisciplinary Major of Ocean Renewable Energy Engineering, National Korea Maritime & Ocean University, 727 Taejong-ro, Yeongdo-gu, Busan 49112, Korea; meeju9988@naver.com

**Keywords:** concrete median barrier, gradient boosting machine, smoothed particle hydrodynamics, deep neural network, artificial neural network, fragments, travel distance

## Abstract

In the present study, the amount of fragments generated and their travel distances due to vehicle collision with concrete median barrier (CMB) was analyzed and predicted. In this regard, machine learning was applied to the results of numerical analysis, which were developed by comparing with field test. The numerical model was developed using smoothed particle hydrodynamics (SPH). SPH is a mesh-free method that can be used to predict the amount of fragments and their travel distances from concrete structures under impact loading. In addition, deep neural network (DNN) and gradient boosting machine (GBM) were also employed as machine learning methods. In this study, the results of DNN, GBM, and numerical analysis were then compared with the conducted field test. Such comparisons revealed that numerical analysis generated lower error than both DNN and GBM. When prediction results of both the amount of fragments and their travel distances were considered, the result of DNN showed smaller errors than that of GBM. Therefore, in studies where machine learning is used to predict the amount of fragments and their travel distances, careful selection of an appropriate method from the various available machine learning methods such as DNN, GBM, and random forest is absolutely important.

## 1. Introduction

In Korea, to ensure passenger protection, the structural performance of concrete median barriers on expressways are evaluated periodically through certain tests [1]. In the passenger protection test, the theoretical head impact velocity (THIV) and postimpact head deceleration (PHD) are limited to 33 km/h and 30 G (gravity acceleration), respectively. In the structural performance test, the weight and travel distance of the fragments of concrete median barriers should not exceed 2 kg and 2 m, respectively. Furthermore, the structural performance and occupant protection performance are affected by uncertainty factors, such as strain rate effect, size effect, characteristics of the impactor, and heterogeneous material.

Figure 1 shows fragments generated from a secondary accident caused by a damaged fender installed on top of a concrete median barrier (CMB) due to collision of a truck with the CMB. It is already known that due to certain uncertainties, prediction of the amount of fragments generated and their travel distances is not always accurate. For example, Kim et al. [2] reported that under the same impact energy, the amount of fragments and their travel distances can vary depending on the velocity and mass of the impactors.

According to Chopra et al., inaccurate input values can also make prediction difficult [3]. In such scenarios where there are uncertainties and inaccurate information, the movement of fragments after a collision can be predicted by machine learning effectively.

However, a large amount of data is required in machine learning for training, and it is time-consuming to gather such a large amount of date through actual experiments. To this end, finite element method (FEM) has been employed to increase the training data [2,4,5]. Through FEM, any cracks in a concrete structure are simulated by deleting elements, as reported by Lee and Kim et al. [6,7,8,9]. However, such element deletion is not applicable in the current study, as it would affect the amount of fragments. Instead, smoothed particle hydrodynamics (SPH), a mesh-free method developed by Rabczuk and Eibl [10], would be a suitable technique to predict the amount of fragments and their travel distances without any loss of concrete weight.

Furthermore, there has been no parametric study reported to-date on the prediction of concrete fragments and their travel distance. Therefore, in the present study, we developed an analytical model using SPH to predict structural performance of median barriers, which depends upon certain major variables viz. concrete thickness, compressive strength, reinforcement ratio, and collision conditions. The newly developed model was also evaluated by comparing the model results with that of a field test. Furthermore, prediction studies using machine learning have been conducted in various areas [11,12,13,14,15]. However, there has been no parametric study on the prediction of concrete fragments and their travel distance. Therefore, using the analytical results as training data for machine learning, it was possible to accurately predict the amount of fragments generated after a collision and their travel distances.

The study was conducted in 3 steps. The first step was numerical model development and verification (Section 3.1, Section 3.2 and Section 3.3). The second step was construction of DNN and LightGBM to predict concrete fragments and their travel distance based on the developed numerical model (Section 4.1, Section 4.2, Section 4.3 and Section 4.4). The third and final step was a comparative study of the results of DNN and LightGBM with that of field test (Section 4.5).

## 2. Smoothed Particle Hydrodynamics Model to Predict the Amount of Fragments and Travel Distance

In this section, theoretical backgrounds of SPH and a concrete material model are discussed. The concrete material model used here is called the continuous surface cap (CSC) model [16,17]. This roadside safety material model was developed by the Federal Highway Administration (FHWA) of the United States to analyze collisions between vehicles and roadside facilities.

### 2.1. Theoretical Background of SPH

The smoothed particle method was developed to solve astrophysical problems. The SPH is a mesh-free method based on Lagrangian formulation. Therefore, SPH does not involve locking of elements or negative volume and can simulate large deformations. For this reason, the SPH method has found wide applications, e.g., in fluid mechanics and soil mechanics. Benz and Asphaug [18] simulated fractures in brittle solids under impact loadings using the SPH method. Liu et al. [19,20] performed computer simulation of large explosions using SPH. The SPH method typically considers interactions of each particle as field functions and their differential form, and the approximations made in this method can be expressed as follows:(1)$$f\left(x\right)=\overset{N}{\underset{j=1}{\sum }}\frac{m_{j}}{\rho_{j}}f\left(x_{j}\right)W\left(x_{i}-x_{j},h\right)$$
(2)$$\nabla \cdot f\left(x\right)=-\overset{N}{\underset{j=1}{\sum }}\frac{m_{j}}{\rho_{j}}f\left(x_{j}\right)\cdot \nabla W\left(x-x_{j},h\right)$$
where $f\left(x\right)$ are field functions, $\nabla \cdot f\left(x\right)$ is differential form, *i* is particle, $m_{j}$ is mass, $\rho_{j}$ is density, *h* is smoothing length, $W\left(x_{i}-x_{j},h\right)$ is the value of the kernel function, and N is the total number of particles. ∇W in the above equation is the gradient, which is evaluated at particle j. A detailed explanation of the SPH method can be found in Liu and Liu [20].

### 2.2. Theoretical Background of Continuous Surface Cap (CSC) Model

The CSC model has several capabilities, such as multiaxial strength, stiffness degradation and dilation, and strain rate effects. The CSC model can simulate continuous intersection between the failure surface and hardening cap. The shear failure surface of this model has a shape of affine-exponential spine and is expressed as
(3)$$F_{f}\left(J_{1}\right)=\alpha_{1}-\lambda_{1}\exp^{\left(\beta_{1}J_{1}\right)}+\theta_{1}J_{1}$$
where, α, β, λ, and θ are values selected by fitting the model surface to strength measurements from a triaxial compression test of plain concrete cylinders.

Fracture energy is an important key parameter to determine generation of cracks due to tension. In the present study, the fracture energy was estimated based on fib Model code 2010 [21], as shown in Equations (4) and (5):(4)$$f_{ctm}=0.3{\left(f_{ck}\right)}^{2/3}\;$$
(5)$$G_{F}=73•f_{cm}^{0.18}$$
where f_ctm_ is tensile strength, f_ck_ is characteristic compressive strength, f_cm_ is mean compressive strength, and G_F_ is the fracture energy of concrete.

## 3. Introduction of the Developed Local Impact Model

According to Kim et al. [2], the maximum amount of fragments is generated when top of a median barrier and corner of cargo of a vehicle contact during a collision. The vehicle model of National Crash Analysis Center (NCAC) based on the European standard EN-1317 [22] is generally used in vehicle collision analysis. However, this vehicle model does not simulate local collision between the lower corner zone of a cargo compartment and the upper zone of a concrete median barrier. Therefore, a detailed numerical model was developed in the current work to simulate a local collision.

### 3.1. The Developed Numerical Analysis Model

Figure 2 shows our newly developed numerical analysis model for verification. In this model, the concrete for numerical model was SPH and the wire-mesh for numerical model was beam element. The concrete material model used here is the CSC model [16,17], while the wire-mesh material model used is piecewise linear plasticity, which is based on elastoplastic material model.

The key parameters selected for the CSC model were “tension softening parameter”, “compression softening parameter”, “fracture energy”, “repow”, and “erode”. To estimate the concrete fracture energy, equations from the fib Model code 2020 [21] were used. The “erode” determines the deletion of elements with plastic strain. If an element lacks the material capacity given by its yield surface, the element should be removed from the simulation of the behavior of concrete. Thai et al., El-tawil et al., and Murray selected 1.4, 1.4, and 1.1 for “erode”, respectively [16,23,24]. However, in the current study, 1.0 was selected for “erode” following Kim et al. [2].

Another key parameter, “repow”, is for rate effect parameter. This parameter is used to increase fracture energy based on the rate effect parameter [16,17]. In the present study, the default value of “repow” was selected as 1.0. The selected parameters for tension softening, compression softening, and erode were 0.1, 100, and 1.0, respectively. A detailed discussion of the parameter can be found in Kim et al. [2].

To consider the strain rate effect of wire-mesh, Cowper–Symonds’ equation was used with C and p coefficients. The values of the selected C and p coefficients were 1.05 × 10^7^ and 8.3 according to Chung et al. [25].

The new model was developed by comparing our results with that of Xiao et al. [26]. The original shape of a concrete median barrier is thicker toward the lower end. However, in the new model, a slab-type concrete structure was selected since the lower part of the concrete median barrier does not affect the amount of fragments after collision. The size of the concrete structure selected was 3000 mm × 1270 mm × depth parameter.

The crack patterns and damaged area obtained from the developed numerical model were compared with the experimental results. Such comparison indicated that the developed numerical model could predict crack patterns and damaged area correctly. A detailed discussion of the numerical model can be found in Kim et al. [2].

### 3.2. Research Scopes of CMB

To predict the amount of fragments and travel distance, the scopes of the key parameters need to be established. With limited statistical data of CMB-vehicle collision, research scopes were selected for concrete thickness, compressive strength, reinforcement ratio, impact location, and impact energy (impact velocity and impact mass). However, the impact mass could not be predicted from the results of field test; therefore, we conducted reverse analysis with various masses using the numerical model to estimate the impact mass relevant to local impact of a truck with a CMB, as shown in Figure 3.

Furthermore, impact locations and impact velocities were obtained from recorded video data obtained from the Korea Expressway Corporation [27,28]. The damaged zones of the CMB in both the front and rear side faces were compared with the field test data [6,7,8,29]. A detailed discussion on model verification can be found in Kim et al. [2]. Table 1 shows the scope of the key parameters for numerical analysis.

### 3.3. Prediction and Verification of the Travel Distance

The travel distance was predicted using the initial velocity of the fragments. The prediction equation for fall time is expressed as
(6)$$\frac{1}{2}at^{2}+V_{yi}t-1270=0$$
where *a* is gravitational acceleration (9810 mm/s), *t* is fall time (s), *V_yi_* is the initial velocity in the direction of gravity (mm/s), and 1270 mm is the height of the concrete median barrier.

The prediction equation can be expressed as
(7)$$V_{xi}\times t=D_{L}$$
where, *V_xi_* is the initial lateral velocity (mm/s), $D_{L}$ is the prediction travel distance value (mm), and *t* is the obtained time from Equation (6).

## 4. Application of Machine Learning to Evaluate Structural Performance under Impact

To predict the amount of concrete fragments, Kim et al. [2] conducted multiple linear regression analysis (MRA) considering the numerical analysis results. However, the coefficient of determination (R^2^) was not high since the MRA defines the relationship between independent and dependent variables. Therefore, ANN was employed to improve the relationship between the independent and dependent variables. There are various algorithms based on ANN such as Deep Neural Network, Recurrent Neural Network (RNN), Gated Recurrent Unit (GPU), and Long Short-Term Memory (LSTM). With a computer having high processing speed and sufficient amount of training data available, DNN can be applied very rapidly and effectively in various fields. On the other hand, LightGBM has the advantages of excellent accuracy and fast training processing speed using the leafwise method. Therefore, to predict the amount of fragments and travel distance of concrete under impact loadings, both DNN and LightGBM were constructed based on the results of SPH analysis, taking the uncertainties into account.

### 4.1. Development of ANN

Deep learning was perceived and developed in the way the human brain works. Deep learning progresses typically via three layers, viz., the input layer, hidden layer, and output layer. The results from the input layer are combined in the hidden layer using a combination function. The results are then weighted, calculated, and sent to the output layer via an activation function. In the current work, to minimize error, a backpropagation technique was followed in which the errors found in the output layer were calculated backward (see Figure 4).

There are many activation functions, such as sigmoid, rectified linear unit (ReLU), and hyperbolic tangent function. Among these, the ReLU is used most frequently and we used this in the current study.

Abbas and Jang performed future data prediction using ReLU and tanh as activation functions, with Adam and RMSprop as optimizers [11]. Results indicated that ReLU and Adam showed better performances than tanh. Therefore, ReLu and Adam were selected as the activation function and optimizer, respectively, in this study.

For learning rate, we selected the values that would not generate underfitting or overfitting by considering combinations with other parameters using the values, 0.01, 0.001, and 0.0001. Generally, large epoch values are needed for small training rates, and small epoch values for large training rates. The caveat here is that optimum weight values are not determined when large training rates are used, which results in inaccurate prediction results.

In the present work, the DNN was optimized considering the number of layers, the number of nodes, and epoch so that the DNN does not generate overfitting or underfitting. Table 2 shows the selected parameters.

### 4.2. Development of Gradient Boosting Method

The boosting method is one of the ensemble methods that utilizes decision trees, as shown in Figure 5. It continuously improves an error from a single model. It generally exhibits high accuracy because it ultimately applies all training models considered so far. There can be outliers, which lie outside the normal distribution, in the distribution of each variable.

Furthermore, to note, where the boosting method assigns one weight to all the models, the gradient boosting machine (GBM) assigns different weights to an individual model. In GBM, the weights are applied in the form of differentiated errors. The errors in GBM are calculated using the down gradient method, in which the differentiated value becomes the minimum. The Loss function of GBM can be represented as
(8)$$j\left(y_{i},\;f\left(x_{i}\right)\right)=\frac{1}{2}{\left(y_{i}-f\left(x_{i}\right)\right)}^{2}$$
where $y_{i}$ are the observed values and $f\left(x_{i}\right)$ are the predicted values. The gradient can be represented as shown below.
(9)$$\frac{\partial j\left(y_{i},f\left(x_{i}\right)\right)}{\partial f\left(x_{i}\right)}=\frac{\partial \left[\frac{1}{2}{\left(y_{i}-f\left(x_{i}\right)\right)}^{2}\right]}{\partial f\left(x_{i}\right)}=f\left(x_{i}\right)-y_{i}$$
where the most optimum decision tree is made to predict the minimized residual via repeated iteration.

Training is carried out in such a way so that it complements the error of the previous tree. To minimize error, a second model is made to predict the error of the first model, and a third model is made to predict the error of the second model. This iteration is repeated to create a final model that reduces the error and can be used to make predictions. The advantage of GBM is that the training and prediction times are very fast because it can create an asymmetric tree shape. The accuracy of this method is also known to be very good.

In the present study, we carried out GBM by considering the learning rate, the number of estimators, and the number of iterations using LightGBM developed by Microsoft in 2016. The boosting algorithm horizontally expands the tree by level with levelwise method. On the other hand, LightGBM is an algorithm that vertically expands the tree based on leaves with max delta loss through leafwise analysis. Assuming that both algorithms generate the same leaves, we obtained a classification model that is faster and less lossy than the time-consuming levelwise analysis algorithm by scaling symmetrically regardless of the loss rate.

### 4.3. Results of DNN and GBM

In Figure 6, the prediction and numerical analyses results of the amount of fragments and their travel distances using DNN are compared. The coefficients of determination (R^2^) for the amount of fragments and their travel distances are 0.9406 and 0.9165. This indicates that the prediction results of the amount of fragments are better than the analysis results. The prediction results of the travel distances were obtained using the initial velocity and not from the direct analysis results, which we believe is the reason why the prediction results were not accurate. Nevertheless, the coefficient of determination for the prediction of travel distances was as good as 0.9165, and the analysis result is properly reflected.

In Figure 7, the prediction results of the amount of fragments using GBM and the analysis results are compared. The coefficient of determination for the amount of fragments was 0.9795, while the coefficient for their travel distances was 0.9207. Similar to the prediction results obtained using DNN, the prediction results of the amount of fragments by GBM are better than the analysis results. Furthermore, the coefficients of determination obtained using GBM are higher than those obtained using DNN. We can thus confirm that the prediction results of GBM are better than the prediction results of the analysis.

The degrees of importance of the variables in predicting the amount of fragments and their travel distances using GBM are shown in Figure 8 and Figure 9. It was observed that the amount of fragments was affected mostly by concrete thickness and the travel distance was most affected by collision speed. On the other hand, the amount of fragments and their travel distances were affected least by compressive strength of the concrete and the reinforcement ratio. Accordingly, the compressive strength in the range 25.5–34.5 MPa used in this study did not affect the amount of fragments and their travel distances. Figure 9 shows the decision tree structure for predicting the amount of concrete fragments considering various parameters. The selected parameters were learning rate 0.01, number of iterations 20,000, and number of estimators 1000.

Table 2 shows the final mean absolute error (MAE) and R^2^ after the training using DNN and GBM. MAE was smaller and R^2^ was larger in GBM than in DNN. The error between the prediction and the analysis results was smaller in DNN, while performance of the regression model was better in GBM. Generally, high MAE values tend to show low R^2^ values. However, the reason for high R^2^ values at such low MAE values is because the same training data were not used. Only 80% of the training data were randomly selected to carry out the training.

### 4.4. Results of the DNN and Gradient Boosting Machines

In Figure 10, the prediction results from the test data that were not used in learning are compared with those from the numerical analysis. For the prediction on the amount of fragments, R^2^ for DNN was 0.8759 and R^2^ for GBM was 0.5853, whereas for the prediction on the travel distance, R^2^ for DNN was 0.5308 and R^2^ for GBM was 0.0866. It can be observed that DNN reflected better predictions than the analysis results. When compared with the training data, the R^2^ value was higher in GBM than in DNN. However, for the prediction using the test data, GBM showed a relatively smaller R^2^ value than DNN. Therefore, the GBM models showed the best for the training data, whereas the DNN models showed improved results for both training data and test data. This means that DNN gave better prediction than GBM. The studies by Nawar and Mouazen [30] also showed similar trends. This points out that depending on the training data set, one method may outperform other methods. Therefore, while conducting machine learning studies of prediction, one should evaluate various machine learning techniques, such as DNN, GBM, and Random Forest, and then select a proper technique.

### 4.5. Comparison of the Prediction Results with Experimental Test

A field test was performed as shown in Figure 11. The amount of fragments and travel distances predicted from DNN, GBM, and numerical analyses with the same dimensions and conditions were compared with the field test results. Table 3 shows the MAE and R^2^ of DNN and LightGBM. Table 4 exhibits both the test results and prediction values, whereas Figure 12 shows the relative accuracies with respect to the test results. For machine learning, the amount of fragments predicted by DNN and GBM were 20 kg and 14 kg, respectively. The errors with respect to the real test were 23.1% and 46.2%, respectively. By contrast, the amount of fragments predicted by numerical analysis was 33 kg, and the difference from the real test data was 26.9%. Therefore, it can be seen that DNN exhibited the smallest error. The values of travel distance obtained from DNN, GBM, and numerical analysis were 456 mm (30.9%), 297 mm (55.0%), and 815 mm (23.5%). When the amount of fragments and their travel distances were considered jointly, the average error for numerical analysis, DNN, and GBM were 25.2%, 27.0%, and 50.6%, respectively. Therefore, for both fragmentation amount and travel distance, numerical analysis generated results closer to the field test ones than DNN and GBM. In machine learning, the DNN showed smaller errors than GBM with respect to the field test results. Therefore, it can be inferred that DNN was more efficient than GBM in predicting the amount of fragments and their travel distances after collision with concrete structures and various input variables, nonlinearities, and uncertainties were inherent in the analysis.

To note, in South Korea, if the travel distance and amount of fragments are more than 2 m and 2 kg, respectively, it is considered a failed test. The CMB in such a case does not meet the real impact test guideline for vehicle safety guard [1] and is considered to have poor performance. When these regulations were applied, it was found that all the methodologies employed in this study, i.e., the numerical analysis, DNN, and GBM—passed the test and met those guidelines. Furthermore, the field test results also showed that all the guidelines were fulfilled (no more than 2 kg and 2 m). Therefore, if machine learning is learned based on pass/fail, it is expected that the accuracy of prediction will improve.

## 5. Conclusions and Future Study

In the current study, to predict the amount of fragments and their travel distances, a collision was analyzed and the results were predicted using SPH. The analytical model closely simulated the configurations of the fracture and the scope of damage to the concrete after collision.

Using the newly developed analytical model, various variables were analyzed and the results were utilized as training data for DNN and GBM. The results of the training showed that MAE was smaller in DNN, and R^2^ was larger in GBM. Therefore, the error in the analysis results was smaller in DNN, and the performance of the regression model was higher in GBM. When the field test results were compared with the results of the numerical analysis, DNN, and GBM, the average errors for the amount of fragments and their travel distances were 25.2%, 27%, and 50.6% for the numerical analysis, DNN, and GBM, respectively.

From the results of numerical analysis, DNN, and GBM prediction, it was found that no fragments and travel distance values obtained were more than 2 kg and 2 m, respectively. In the field test, neither the fragments amount nor the travel distance were more than 2 kg and 2 m, respectively. The developed numerical model, DNN, and GBM were found to predict the field test results very well following the real impact test guidelines for vehicle safety guard [1].

In the study of fragments amount and travel distance prediction using machine learning, the prediction results showed different errors depending on the method of machine learning. Therefore, from the various machine learning methods, such as DNN, GBM, and random forest, it is important to select an appropriate one.

## Figures and Tables

**Figure 1 materials-15-01045-f001:**
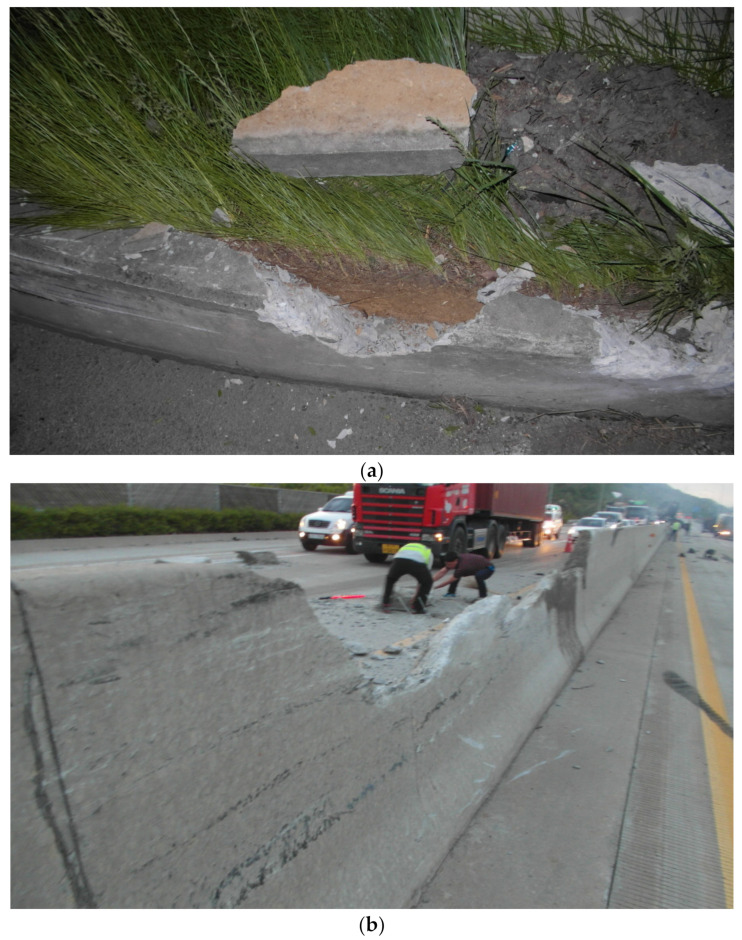
Fragments and damaged shape of concrete median barrier. (**a**) Fragments generated after collision with a truck. (**b**) Damaged shape after collision.

**Figure 2 materials-15-01045-f002:**
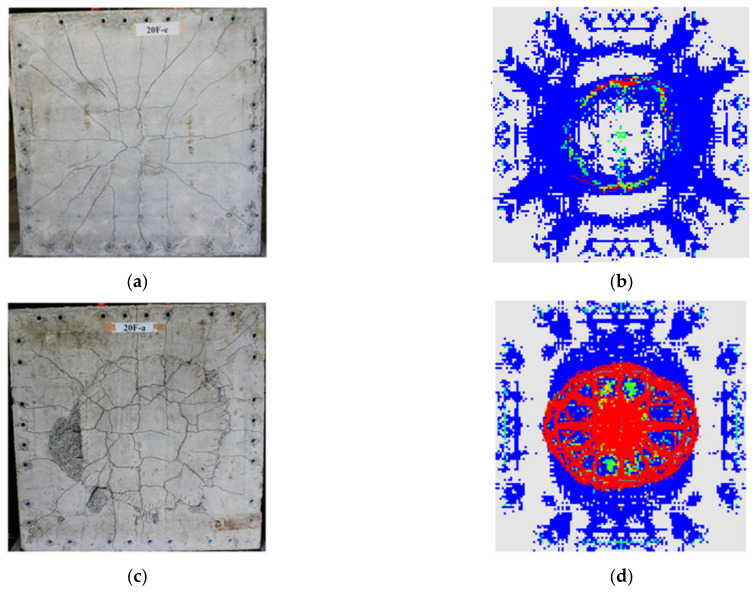
SPH numerical model verification [26]. (**a**) Experimental results (20F-e); (**b**) Analysis results (20F-e); (**c**) Experimental results (20F-a); (**d**) Analysis results (20F-a).

**Figure 3 materials-15-01045-f003:**
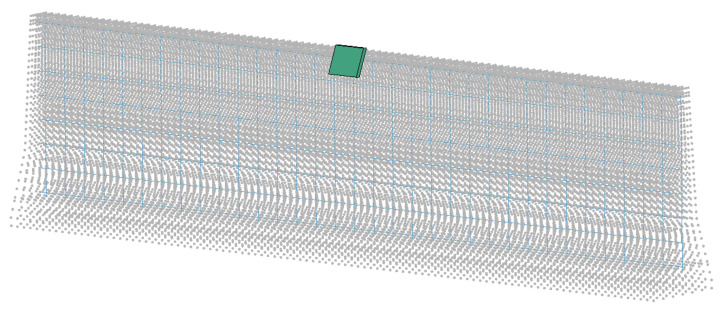
Numerical model for reverse analysis.

**Figure 4 materials-15-01045-f004:**
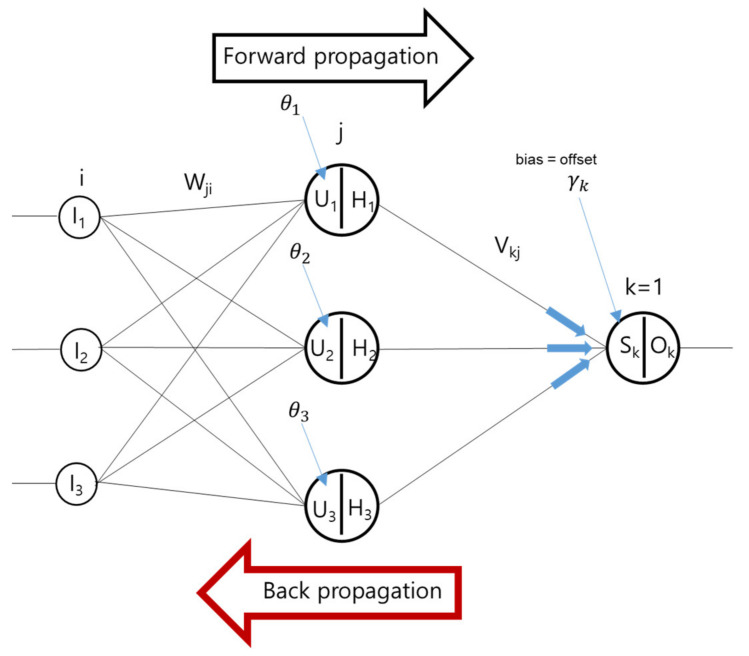
Forward propagation versus backward propagation.

**Figure 5 materials-15-01045-f005:**
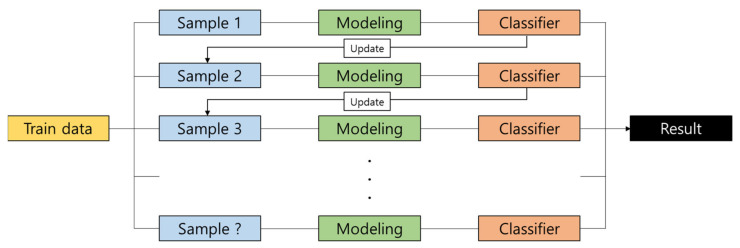
Boosting algorithm.

**Figure 6 materials-15-01045-f006:**
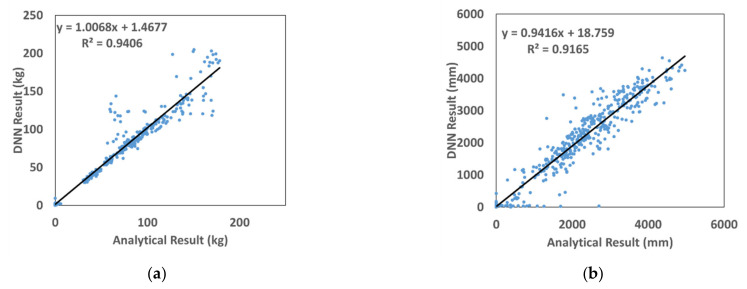
Comparison of DNN results with SPH ones. (**a**) Fragmentation. (**b**) Travel distance.

**Figure 7 materials-15-01045-f007:**
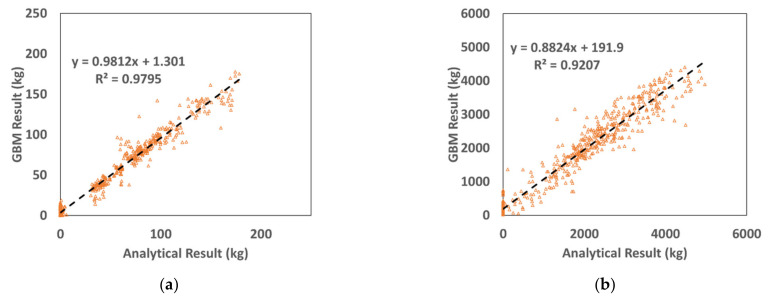
Comparison of the GBM results with SPH results. (**a**) Fragmentation. (**b**) Travel distance.

**Figure 8 materials-15-01045-f008:**
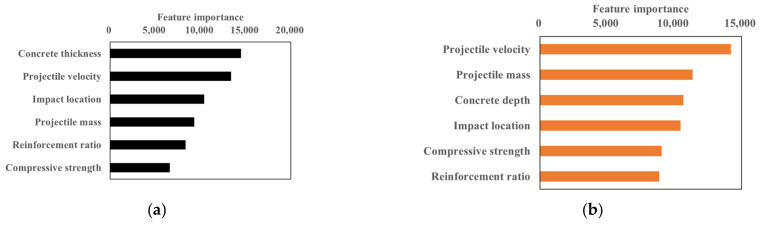
Importance of features of fragments and their travel distance. (**a**) Fragments. (**b**) Travel distance.

**Figure 9 materials-15-01045-f009:**
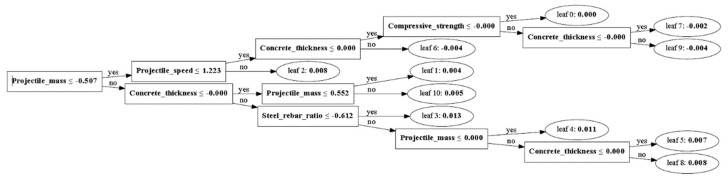
Visualization of the decision tree (fragments).

**Figure 10 materials-15-01045-f010:**
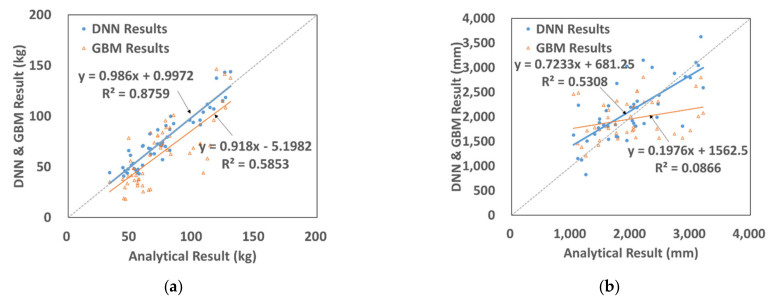
Comparison of the DNN and GBM results in regard to fragmentation amount and travel distance. (**a**) Fragmentation. (**b**) Travel distance.

**Figure 11 materials-15-01045-f011:**
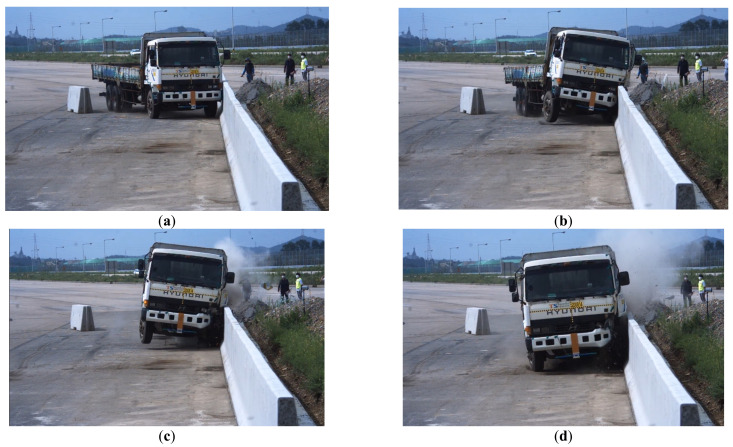
Field test. (**a**) 0.00 s; (**b**) 0.03 s; (**c**) 0.07 s; (**d**) 0.10 s.

**Figure 12 materials-15-01045-f012:**
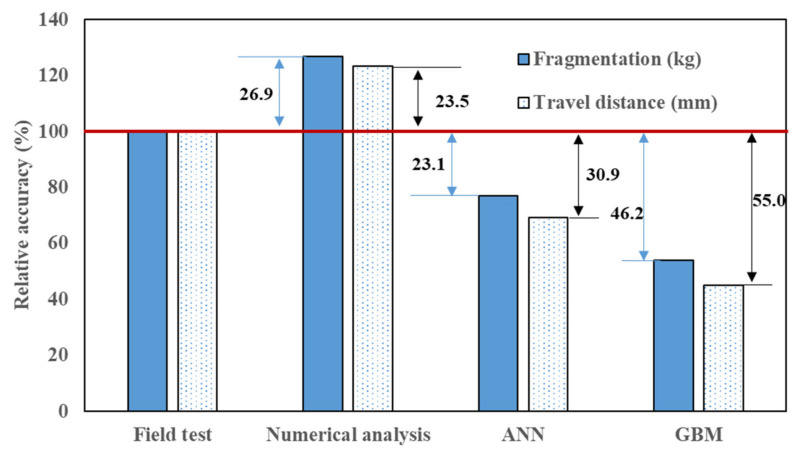
Comparison of different prediction results.

**Table 1 materials-15-01045-t001:** Scope of the key parameters.

Parameters				Minimum	Maximum
Concrete	Concrete compressive strength			25.5 MPa	34.5 MPa
	Concrete thickness			150 mm	250 mm
Reinforcement	Reinforcement ratio			0.0	0.4
Impactor	Impact location from the top surface			80 mm	140 mm
	Impact energy	3.2 kJ	Velocity	17.0 km/h	22.8 km/h
			Mass	160 kg	280 kg
		10.8 kJ	Velocity	17.0 km/h	36.0 km/h
			Mass	210 kg	970 kg
		18.0 kJ	Velocity	17.0 km/h	36.0 km/h
			Mass	360 kg	1600 kg

**Table 2 materials-15-01045-t002:** Selected parameters.

Factor	Selected Parameter
Learning rate	0.001
Epoch	2000
Number of layer	3
Number of node	32, 16, 8
Activation function	ReLU
Weight adjustment	Stochastic Gradient Descent
Optimizer	ADAM

**Table 3 materials-15-01045-t003:** The results of DNN and GBM.

	DNN		LightGBM	
	Fragmentation	Travel Distance	Fragmentation	Travel Distance
MAE	3.4848	338.2805	7.7368	399.756
R^2^	0.9406	0.9165	0.9795	0.9207

**Table 4 materials-15-01045-t004:** Comparison of different prediction results.

	Field Test	Numerical Analysis		DNN		LightGBM	
	Measured Value	Predicted Value	Error	Predicted Value	Error	Predicted Value	Error
Fragmentation	26 kg	33 kg	26.9%	20 kg	23.1%	14 kg	46.2%
Travel distance	660 mm	815 mm	23.5%	456 mm	30.9%	297 mm	55.0%

## Data Availability

Not applicable.

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
