# Peer review of "Prediction of Concrete Fragments Amount and Travel Distance under Impact Loading Using Deep Neural Network and Gradient Boosting Method"

_materials, 2022, doi:10.3390/ma15031045_

Round 1
Reviewer 1 Report
This paper uses a machine learning method to predict the number and movement distance of debris generated by CMB after collision with vehicles, which has a certain workload and innovation. In summary, the research is interesting and provides valuable results, but the current document has several weaknesses that must be strengthened to obtain a documentary result that is equal to the value of the publication.
- Why does the author choose the traditional machine learning algorithm rather than the current popular deep learning algorithm?
- The research gap and current novel algorithms are not well presented.
- The document contains a total of 25 employed references only, the total number is insufficient, and their actuality is not quite More 2021-2022 papers should be referred to.
- Would you introduce this study's development prospects and application scenarios?
- There are many minor typographical errors in this article, and the author might adjust the layout to enhance readability, such as Figure 11 and Formula Numbers.
- Since there are few papers cited in this paper, the author is encouraged to quote more latest articles to improve the integrity of the introduction and the credibility of the article. For the algorithm of predicting concrete properties by machine learning method, please refer to doi.org/10.1016/j.conbuildmat.2021.125970
Reviewer 2 Report
The paper proposes a method to predict concrete fragments and travel distance under impact loadings using artificial neural networks and the gradient boosting method. The paper fits to the journal. The introduction is well structured. The state of the art is concise yet sufficient to deduce the research gap. The method is appropriately introduced, including sufficient background information, and the research design is appropriate. The validation (field) tests are comprehensible and the conclusions are supported by the results.
The major problem of this paper is related to English writing and formal appearance, which compromise the paper quality, render the paper hard to read, and require substantial improvement. Deficiencies are obvious in article usage along with the grammatical number. For example (as one example out of many), “theoretical background of CSC model” requires either an article, which would be grammatically correct but semantically questionable, or the use of plural form. Moreover, emotive and subjective language must be avoided in academic writing (e.g. “…it is not easy…”). Acronyms alone (i.e. without full term) must be avoided in the keyword list and in the headings (e.g. “SPH model…”). Regarding the formal appearance, variables must be italicized in the text and in the equations (e.g. 117, 235, and many more). Capital and small letters must not be mixed (e.g. “The Developed numerical analysis model”, 126). In general, adjectives are written in small letters, only proper nouns need to be capitalized. For example, “sigmoid” not “Sigmoid” (203). The grammatical tense must be consistent (e.g. section 4.1, “...the results ARE weighted … back propagation WAS utilized…”). Many figures, such as Figure 9, are hardly comprehensible because of the poor quality and the small font size. Contracted forms must be avoided in academic writing (e.g. 328).
Regarding references 6,7,8,9 in line 59, these can be reduced to one reference. The studies are not elucidated content-wise and there is no added value of the extensive self-citations in this particular context.
Round 2
Reviewer 1 Report
Congrats!
The authors have successfully addressed all my comments. Therefore, I recommend the publication of this manuscript.
Reviewer 2 Report
The paper has been modified appropriately and is ready for publication.